# Adzuki Bean MY59 Extract Reduces Insulin Resistance and Hepatic Steatosis in High-Fat-Fed Mice via the Downregulation of Lipocalin-2

**DOI:** 10.3390/nu14235049

**Published:** 2022-11-27

**Authors:** Jaewoong Lee, Byong Won Lee, Kyung Eun Kim, Hyeong Seok An, Eun Ae Jeong, Hyun Joo Shin, Seok Bo Song, Gu Seob Roh

**Affiliations:** 1Department of Anatomy and Convergence Medical Science, College of Medicine, Institute of Health Sciences, Gyeongsang National University, Jinju 52727, Republic of Korea; 2Department of Southern Area Crop Science, National Institute of Crop Science, Rural Development Administration, Miryang 50424, Republic of Korea

**Keywords:** adzuki bean extract, lipocalin-2, insulin resistance, hepatic steatosis

## Abstract

Adzuki bean is well known as a potential functional food that improves metabolic complications from obesity and diabetes. Lipocalin-2 (LCN2) has been implicated to have an important role in obesity and diabetes. However, the protective roles of adzuki bean MY59 extract (ABE) on insulin resistance and hepatic steatosis are not fully understood. In the present study, we investigated the effects of ABE on LCN2 expression in high-fat diet (HFD)-fed mice. ABE reduced HFD-induced fat mass and improved insulin resistance. In addition to hepatic steatosis, HFD-fed mice showed many apoptotic cells and neutrophils in the epididymal fat pads. However, these findings were significantly reduced by ABE supplementation. In particular, we found that increased LCN2 proteins from serum, epididymal fat pads, and liver in HFD-fed mice are significantly reduced by ABE. Furthermore, ABE reduced increased heme oxygenase-1 and superoxide dismutase-1 expressions in adipose tissue and liver in HFD-fed mice. We found that hepatic nuclear factor-kappa B (NF-κB) p65 expression in HFD-fed mice was also reduced by ABE. Thus, these findings indicate that ABE feeding could improve insulin resistance and hepatic steatosis by decreasing LCN2-mediated inflammation and oxidative stress in HFD-fed mice.

## 1. Introduction

Obesity is a worldwide epidemic health problem that causes metabolic dysfunction, including non-alcoholic fatty liver disease (NAFLD), cardiovascular disease, and type 2 diabetes mellitus [1]. This metabolic dysfunction in obesity is closely related to increasing pro-inflammatory adipocytokine [2,3].

Lipocalin 2 (LCN2) is a pro-inflammatory adipocytokine and is mainly expressed in various organs, including adipose tissue, liver, heart, and kidney [4,5]. Many studies have demonstrated that high plasma levels and expression for LCN2 were closely associated with obesity-induced metabolic complications such as NAFLD and diabetes [6,7,8]. Moreover, LCN2 deletion improved insulin sensitivity in adipose tissue of diet-induced obesity [9] or attenuated non-alcoholic steatohepatitis (NASH) progression in the liver of methionine- and choline-deficient-diet mice [10]. In particular, proinflammatory LCN2 is regulated by a transcription factor NF-κBp65 [8,11,12], so it has been suggested that LCN2 levels can potentially be a biomarker for obesity-induced metabolic disorders [4].

Adzuki bean, a member of the Fabaceae family, is mainly cultivated and consumed in Asia, including Korea, China, and Japan [13]. Because the nutritional components of adzuki bean consist of 65% carbohydrates and 20% vitamin B, oils, proteins, and minerals, it has been known as one of the nutritionally functional foods [14,15,16]. Some studies have shown that adzuki bean extract improves metabolic diseases such as obesity, diabetes, and fatty liver diseases [13,17,18]. Because adzuki bean contains an abundant content of α-glucosidase inhibitor, which inhibits glucose absorption in the intestine, its extract has been shown to ameliorate the diabetic phenotype in rodents [19,20]. However, although adzuki bean extract has a potential therapeutic agent for obesity-induced metabolic disorders, it is not fully understood.

Here, we investigated the role of adzuki bean extract on insulin resistance and hepatic steatosis in HFD-induced obese mice. Furthermore, we evaluated the anti-inflammatory and antioxidant effects of adzuki bean extract through LCN2-related inflammation in HFD-fed mice.

## 2. Materials and Methods

### 2.1. Plant Material

Adzuki bean MY59 was developed by the National Institute of Crop Science at Rural Development Administration, Republic of Kore in 2013. For cultivation, Chungju-pat (red bean) and *Vigna nakashimae* (wild red bean), which have excellent α-glucosidase inhibitory activity, were crossed, and the pedigree breeding method was applied. Adzuki bean MY59 used in this study was grown at the National Institute of Crop Science, Rural Development Administration, Miryang, Republic of Kore, during the 2020 growing season and stored at 4 °C.

### 2.2. Preparation of the Adzuki Bean MY59 Extract (ABE)

Dried seeds of adzuki bean MY59 (1 kg) were ground in a grinder, extracted with 20 L of 80% ethanol for 3 days while being vigorously shaken at room temperature, and then filtered. The 80% ethanol extract was concentrated in a vacuum under 40 °C to remove ethanol. The remaining water layer was freeze-dried to give 114 g brownish solid. Brownish solid (100 g) was re-extracted (with shaking) with 2 L isopropanol for 24 h at room temperature and then filtered. The extract was concentrated in a vacuum under 40 °C and then freeze-dried to give a crude extract, which was used for biological activities.

### 2.3. Assay for α-Glucosidase Inhibitory Activity

According to Ryu et al. [21], the modified method was used to measure the α-Glucosidase inhibitory activity. An amount of130 μL of potassium phosphate buffer (50 mM, pH 6.8), 25 μL of 2.5 mM p-nitrophenyl-α-D-glucopyranoside, and 20 μL of tested samples in methanol with 10% DMSO were mixed at room temperature for 10 min. The reaction was started by adding 25 μL of 0.25 U/mL α-glucosidase and was then incubated for 30 min at 37 °C. Then 50 μL of 1 M Na_2_CO_3_ were added to the reaction solution to stop the reaction. The absorbance of samples was measured at 405 nm with a microplate reader. Acarbose was used as the positive control in this study. The inhibition rates (%) = [(OD_control_ − OD_control blank_) − (OD_test_ − OD_test blank_)]/(OD_control_ − OD_control blank_) × 100%. The results are expressed as IC_50_ values.

### 2.4. Animals and Diet Model

All mice were maintained in the animal facility at Gyeongsang National University (GNU). Three-week-old male C57BL/6 mice were purchased from KOATECH (Pyeongtaek, Republic of Kore) and were divided into three groups at random for a low-fat diet (*n* = 9, LFD group, 10% kcal fat, Research Diets, New Brunswick, NJ, USA), a high-fat diet (*n* = 10, HFD group, 45% kcal fat, Research Diets), and HFD supplementation ABE (*n* = 10, HFD+ABE group). Based on our previous study [22], the mice were dosed daily with 200 mg/kg of ABE in HFD feed. The mice were fed for 12 weeks starting at 4 weeks. On a 12-h light/12-h dark cycle, all mice were kept in conditions free of viruses.

### 2.5. Echo-MRI

Mice were subjected to echo-MRI (Whole Body Composition Analyzer, Houston, TX, USA) to measure body fat mass.

### 2.6. Glucose Tolerance Test (GTT) and Insulin Tolerance Test (ITT)

GTT and ITT were performed as previously described [23] using D-glucose (2 g/kg, Sigma-Aldrich, St. Louis, MO, USA) or insulin (0.75 U/kg, Humulin-R, Eli Lilly, Indianapolis, IN, USA). After intraperitoneal injection of D-glucose or insulin, blood samples were obtained from tail vein. The glucose levels from GTT and ITT were determined using an Accu-Chek glucometer (Roche Diagnostics GmbH, Mannheim, Germany).

### 2.7. Metabolic Parameters

After overnight fasting, final body weights and blood glucose was measured, and mice were anesthetized with Zoletil (20 mg/kg, Virbac Laboratories, Carros, France) and Rompun (5 mg/kg, Bayer, Bayer Korea, Republic of Kore). From the left ventricle, blood samples were taken and centrifuged. Serum alanine aminotransferase (ALT) and total cholesterol levels were measured at the Green Cross Reference Laboratory (Youngin-si, Republic of Korea).

### 2.8. Enzyme-Linked Immunosorbent Assay (ELISA)

We determined circulating levels of insulin, leptin, and LCN2 using mouse insulin (Shibayagi Co., Gunma, Japan), leptin (R&D Systems, Minneapolis, MN, USA), and mouse LCN2 (R&D Systems) ELISA kits.

### 2.9. Hepatic Triglyceride (TG) Colorimetric Assay

TG concentrations in frozen livers were measured by a TG colorimetric assay kit (Cayman Chemical Company, Ann Arbor, MI, USA).

### 2.10. Hematoxylin and Eosin (H&E) Staining and NAFLD Activity Score Measurement

For histopathological analysis, the mice were perfused with 4% paraformaldehyde in 0.1 M phosphate-buffered saline (PBS). Tissues, including epididymal fat pads, liver, and pancreas, were fixed in 4% paraformaldehyde for 12 h at 4 °C, embedded in paraffin, and cut into 5-µm sections. The sections were stained with H&E (Sigma-Aldrich) and were visualized under BX51 light microscopy (Olympus, Tokyo, Japan). To determine the NAFLD activity score, we measured the scores of steatosis (0–3), lobular inflammation (0–2), and hepatocellular ballooning (0–2) [24].

### 2.11. Nile Red Staining

Frozen liver sections were stained with Nile Red (Sigma-Aldrich) to identify hepatic lipid accumulation. The sections were visualized under BX51 light microscopy (Olympus), and digital images were captured and documented. The percentage of Nile Red-positive area (250 × 250 µm^2^) in three sections was measured using i-Solution (IMT i-Solution, Inc., Vancouver, BC, Canada).

### 2.12. Terminal Deoxynucleotidyl Transferase dUTP Nick End Labeling (TUNEL) Assay

To measure the degree of apoptosis in the epididymal fat pads, we performed TUNEL analyses using an in-situ cell death detection kit (Roche Molecular Biochemicals, Mannheim, Germany) according to the manufacturer’s protocol.

### 2.13. Double Immunofluorescence

Sections of deparaffinized epididymal fat pads and liver were incubated with 5% serum for 1 h at room temperature followed by incubation with primary antibodies against anti-LCN2 (R&D Systems), anti-myeloperoxidase (MPO, Abcam, Cambridge, MA, USA), and anti-perilipin-1 (Abcam). After washing three times, the sections were incubated with corresponding Alexa Fluor 488- or 594-conjugated secondary antibody (Invitrogen, Carlsbad, CA, USA). Nuclei were counterstained with 4′, 6-Diamidino-2-phenylindole (DAPI, Invitrogen). Immunofluorescence staining for perilipin- was performed after the TUNEL assay in epididymal fat pads. Slides were mounted with VectaMount (Vector Laboratories, Burlingame, CA, USA), and representative images were taken using a BX51-DSU microscope (Olympus). TUNEL-positive cells were counted in the perilipin-1-free region of adipocytes (200 μm × 200 μm) in three sections using ImageJ software (Version 1.52a, NIH, Bethesda, MD, USA). The intensity of LCN2 and MPO co-localization from 6 to 12 fields from 3 mice/group were quantified with ImageJ software (Version 1.52a).

### 2.14. Protein Extraction

For protein extraction, epididymal fat pads and livers were frozen and homogenized in T-PER lysis buffer (Thermo Fisher Scientific, Carlsbad, CA, USA) with a protease and phosphatase inhibitor cocktail (Thermo Fisher Scientific). For nuclear fraction, livers were homogenized in high-salt extraction buffer (20 mM HEPES-KOH; pH 7.9, 1.5 mM MgCl_2_, 420 mM NaCl, 0.2 mM EDTA, 25% glycerol, protease inhibitors, 0.5 mM DTT) and ice-cold lysis buffer (10 mM HEPES-KOH; pH 7.9, 1.5 mM MgCl_2_, 10 mM KCl, protease inhibitors).

### 2.15. Western Blot Analysis

After bicinchoninic acid assay (Thermo Fisher Scientific) for protein concentration, proteins were loaded and electroblotted. The blots were probed with primary antibody against anti-LCN2 (R&D Systems), anti-heme oxygenase-1 (HO-1, StressGen, MI, USA), anti-superoxide dismutase 1 (SOD1, Santa Cruz Biotechnology, CA, USA,), and NF-κBp65 (Cell Signaling, MA, USA). p84 (Abcam), β-actin (Sigma-Aldrich), and α-tubulin (Sigma-Aldrich) were used as internal controls for normalizing protein contents in tissue samples. Protein bands were detected using enhanced chemiluminescence substrates (Pierce, Rockford, IL, USA), and chemiluminescence was analyzed using an LAS-4000 instrument (Fujifilm, Tokyo, Japan). Densitometry analysis was performed using the Multi-Gauge V 3.0 image analysis program (Fujifilm).

### 2.16. Statistical Analyses

One-way analysis of variance (ANOVA), followed by post hoc analysis with Tukey’s test (PRISM 7.0, GraphPad Software Inc., San Diego, CA, USA), was used to determine group differences. Results are presented as the means ± standard of error of the mean (SEM). A *p*-value less than 0.05 was considered significant.

## 3. Results

### 3.1. ABE Has a Strong α-Glucosidase Inhibitory Activity

Although *Vigna nakashimae* has a strong α-glucosidase inhibitory activity, it is difficult to cultivate on a large scale due to its climbing habit. To improve the climbing habit of *Vigna nakashimae*, adzuki bean MY59 was developed by crossing *Vigna nakashimae* with Chungju-pat, which has excellent cultivation characteristics (Figure 1).

Adzuki bean MY59 had improved climbing habit characteristics and had higher a-glucosidase inhibitory c than (IC_50_ = 9.7 ± 0.38 μg/mL) (Table 1). However, when the 80% ethanol extract of the first adzuki bean MY59 was heated to over 80 °C, the α-glucosidase inhibitory activity disappeared. However, when the first extract of adzuki bean MY59 was re-extracted using isopropanol, α-glucosidase inhibitory activity was increased (IC_50_ = 1.9 ± 0.17 μg/mL) and maintained even when heat was applied to 80 °C.

### 3.2. ABE Reduces Fat Mass and Insulin Resistance in HFD-Fed Mice

To examine the anti-obesity effect of ABE on HFD-fed mice, the mice were fed for 12 weeks with HFD supplementing ABE. As shown in Figure 2a, HFD-fed mice exhibited a significant increase of body weight and fat mass compared to LFD-fed mice, whereas only fat mass in HFD-fed mice was significantly reduced by ABE supplementation. In addition to fasting blood glucose levels, HFD-induced glucose tolerance was improved by ABE (Figure 2b,c). However, compared to HFD-fed mice, there was no significant reduction of glucose levels in HFD+ABE-fed mice during ITT (Figure 2c). Hyperinsulinemia and hyperleptinemia in HFD-fed mice were significantly reduced by ABE (Figure 2d). Additionally, histological analysis showed that the Langerhans islet area of β cells is significantly increased in HFD-fed mice, but this area is reversed by ABE (Figure 2e). These results indicate that ABE feeding could improve insulin resistance in HFD-fed mice by reducing fat mass.

### 3.3. ABE Ameliorates Hepatic Steatosis in HFD-Fed Mice

We next investigated the effects of ABE on hepatic steatosis in HFD-fed mice. In line with fat mass changes in HFD-fed mice, ABE significantly reduced HFD-induced liver weight (Figure 3a). As expected, we found that the increased serum ALT and hepatic triglyceride levels in HFD-fed mice were prominently attenuated by ABE supplementation (Figure 3b,c). Histological analysis supported the idea that ABE decreased the accumulations of lipid droplets within hepatocytes in HFD-fed mice (Figure 3d). In accordance with histological findings, increased NAFLD activity score and Nile Red-stained areas in HFD-fed mice were also significantly reduced by ABE (Figure 3e,f). These data indicate that ABE could improve hepatic steatosis in HFD-fed mice.

### 3.4. ABE Inhibits Apoptotic Adipocytes in HFD-Fed Mice

HFD-induced obesity promotes adipocyte death and immune cell infiltration in the adipose tissue, for which they exacerbate inflammation and insulin resistance. Crown-like structures (CLSs) in the adipose tissue are a histological marker of inflammation [2,3]. We assessed the histological analysis in the epididymal fat pads using H&E and TUNEL staining. H&E staining revealed that HFD-induced CLSs were significantly reversed by ABE administration (Figure 4a,b). HFD-fed mice exhibited many more TUNEL-positive cells around perilipin 1-free adipocytes than LFD-fed mice (Figure 4a). However, ABE administration reduced TUNEL-positive adipocytes (Figure 4c). These data suggest that ABE could protect against HFD-induced adipocyte death and improve insulin resistance.

### 3.5. ABE Reduces LCN2, HO-1, and SOD1 Expressions in the Adipose Tissue of HFD-Fed Mice

Increased CLSs and apoptotic adipocytes in HFD-fed mice were closely related to neutrophil infiltration and LCN2 levels [25,26]. Therefore, we measured serum LCN2 levels using ELISA. Compared to LFD-fed mice, serum LCN2 levels were significantly increased in HFD-fed mice. However, increased LCN2 levels were decreased in HFD+ABE-fed mice (Figure 5a). Double immunofluorescence showed that many LCN2-positive cells were co-localized with neutrophil marker MPO-positive cells in the adipose tissue of HFD-fed mice, whereas they were prominently decreased in HFD+ABE-fed mice (Figure 5b). Additionally, we found that increased LCN2 protein levels in the adipose tissue of HFD-fed mice were reduced by ABE (Figure 5c). Given that LCN2 overexpression was associated with oxidative stress [27], we further examined whether ABE affects HO-1 and SOD1 expression in the epididymal fat pads. As expected, ABE supplementation significantly reduced HFD-induced HO-1 and SOD1 expressions (Figure 5c). These data indicate that ABE could reduce LCN2-related inflammation and oxidative stress in the adipose tissue of obese mice.

### 3.6. ABE Reduces Hepatic Inflammation and Oxidative Stress in HFD-Fed Mice

To investigate the protective effects of ABE on inflammation and oxidative stress in hepatic steatosis, we examined hepatic LCN2, HO-1, and SOD1 expressions (Figure 6). In line with LCN2-positive neutrophils in the adipose tissues (Figure 5), we also found that many LCN2-positive cells are observed in MPO-positive neutrophils in the liver sections of HFD-fed mice compared to LFD-fed mice (Figure 6a). However, ABE supplementation significantly attenuated LCN2-positive neutrophils in the liver of HFD-fed mice. Western blot analysis showed that HFD-induced LCN2, HO-1, and SOD1 expressions were significantly decreased in the liver of HFD+ABE-fed mice (Figure 6b). Furthermore, we investigated the effect of ABE on the nuclear expression of NF-kBp65 as a transcription factor of LCN2 [8,12]. The nuclear NF-kBp65 level was reduced in HFD+ABE-fed mice compared to HFD-fed mice (Figure 6c). These results suggest that ABE may play an important role in anti-inflammation and anti-oxidative stress in hepatic steatosis.

## 4. Discussion

Adzuki bean is well-known for preventing lipid accumulation and adipocytokine production and improving glucose tolerance [13]. This study demonstrated that ABE significantly reduces fat mass, insulin resistance, and hepatic steatosis in HFD-fed mice. As well as circulating LCN2 levels, in particular, LCN2-positive neutrophils in the epididymal fat pads and liver in HFD-fed mice were also reduced by ABE administration. Furthermore, ABE attenuated increased oxidative stress in the adipose tissue and liver of HFD-fed mice. Thus, these findings suggest that ABE may be helpful in preventing obesity-induced insulin resistance and hepatic steatosis.

As α-glucosidase inhibitors are known to have protective effects on obesity and diabetes [28], adzuki beans also have a high content of a potent α-glucosidase inhibitor [29]. Adzuki beans containing polyphenols as natural flavonoids attenuated lipid accumulation and improved lipid metabolism in diet-induced obesity. Additionally, it reduced triglyceride accumulation and pro-inflammatory adipocytokines, including interleukin (IL)-6 and monocyte chemoattractant-1, in cultured human adipocytes [13]. Another study has also demonstrated that the extract of adzuki beans reduces blood glucose and the accumulation of hepatic lipids in diabetic rodents [19,20]. Consistent with previous studies [22], we also found that ABE reduced HFD-induced weight gain, insulin resistance, and hepatic steatosis. Additionally, ABE supplementation decreased hepatic lipid accumulation and necrotic adipose tissue in HFD-fed mice. Thus, these results indicate that ABE may be helpful in preventing obesity-related metabolic dysfunction, including insulin resistance and hepatic steatosis.

Obesity-induced lipid accumulation and insulin resistance promote inflammation and cause apparent adipocyte death. This adipocyte death enhances inflammation and immune cell infiltration in HFD-induced obesity. Therefore, the histology of adipose tissue in obesity is characterized by infiltrated immune cells around the necrotic adipocyte, the so-called CLS [2,3,30]. In this study, ABE significantly reduced the number of CLSs and TUNEL-positive cells within perilipin-1-free adipocytes of HFD-induced mice. These data indicate that ABE inhibits adipocyte death and improves insulin resistance in obesity. On the other hand, NAFLD includes the range from hepatic steatosis to nonalcoholic steatohepatitis, fibrosis, and cirrhosis and is closely associated with obesity-related metabolic disorders [31,32]. In particular, abnormal lipid metabolism in obesity elevates hepatic lipid accumulation and promotes inflammation and oxidative stress [33,34]. In this study, we showed a reduction of hepatic lipid accumulation in HFD+ABE-fed mice, suggesting that it prevents hepatic inflammation and oxidative stress. Therefore, this study indicates that ABE may play an important role in the regulation of metabolic inflammation in obesity.

LCN2, a neutrophil gelatinase-associated lipocalin, has been reported to play a critical role in inflammation in obesity [35,36]. Hepatic LCN2 promotes NASH via neutrophil and macrophage crosstalk and increases hepatic inflammation [10]. Thus, because LCN2 plays an important role in the inflammation of adipose tissue and liver from HFD-induced obesity [4], we expected that ABE could reduce LCN2 protein levels in HFD-fed mice. As expected, we found that as the pro-inflammatory adipocytokine, HFD-induced LCN2 production was decreased by ABE and that ABE reduced many LCN2-positive neutrophils in the adipose tissue and liver of HFD-fed mice. Moreover, as with adipose tissue, hepatic inflammation and oxidative stress were decreased by ABE through the downregulation of HO-1 and SOD1. Taken together, these results suggest that ABE may have a protective role in inflammation and oxidative stress in HFD-induced hepatic steatosis by downregulating LCN2 production.

Present studies have demonstrated that LCN2 has an important role, which promotes pro-inflammatory response and oxidative stress in adipose tissue, and the liver [5,27]. The gene expression of LCN2 is well-known to be regulated by NF-kB in pro-inflammatory stimuli, including lipopolysaccharide (LPS), tumor necrosis factor (TNF)-a, and IL-6 [12,37,38,39]. In HFD-induced obesity, the increased gut microbiota dysbiosis enhanced serum LPS contents, thereby promoting systemic inflammation [40,41]. A previous study has demonstrated that the gut microbiota significantly increased the abundance of beneficial bacteria by intake of HFD supplemented with adzuki bean [42]. The authors suggest that adzuki bean supplementation may regulate gut microbiota dysbiosis, which reduces serum LPS contents and lipid metabolic disorders in HFD-induced obesity [42]. Based on this evidence, we found that ABE supplementation inhibited the nuclear localization of NF-kBp65 in the liver of HFD-fed mice. Therefore, our data indicate that ABE may have an anti-inflammatory effect by inhibiting NF-kB activity.

There are limitations to this study. Although we found that ABE improves insulin resistance and hepatic steatosis in HFD-fed mice, we did not directly elucidate the role of LCN2 in inflammation and oxidative stress. In addition, we need to determine which hepatocyte or non-parenchymal cells are affected by ABE. In the future, we will evaluate the protective effects of ABE using in vitro studies, including primary cell culture.

## 5. Conclusions

The present study demonstrates that ABE may be useful in preventing or improving obesity-related metabolic complications via the amelioration of inflammation and oxidative stress. Furthermore, this study may contribute to the elucidation of the regulatory mechanism of adzuki bean as a potential functional food for anti-obesity and anti-diabetes.

## Figures and Tables

**Figure 1 nutrients-14-05049-f001:**
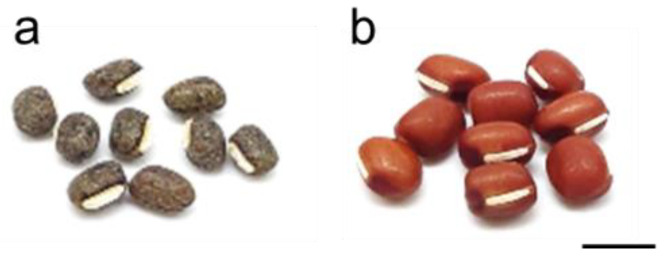
Representative images of the seeds of *Vigna nakashimae* (**a**) and adzuki bean MY59 (**b**). Scale bars, 5 mm.

**Figure 2 nutrients-14-05049-f002:**
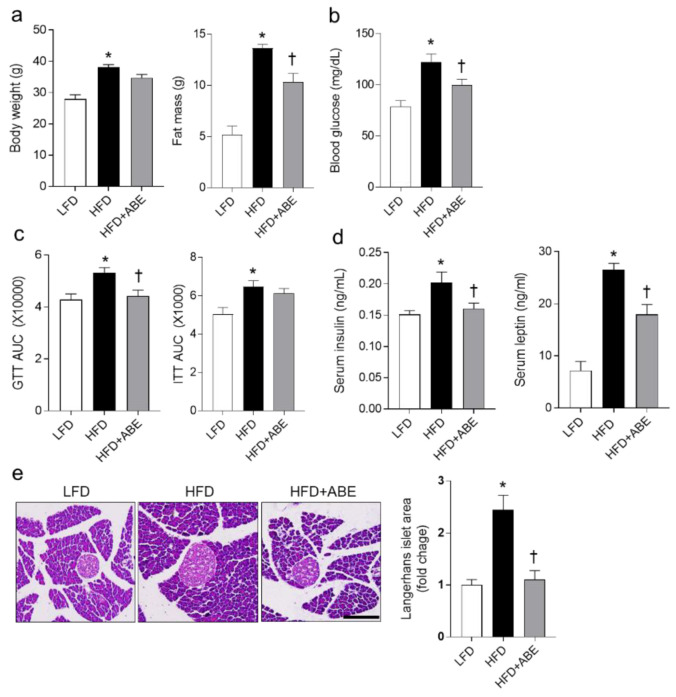
Effects of adzuki bean MY59 extract (ABE) on body weight and insulin resistance in HFD-fed mice. (**a**) Body weight and fat mass. (**b**) Fasting blood glucose level. (**c**) Area under the curve (AUC) for glucose tolerance test (GTT) and insulin tolerance test (ITT). After D-glucose or insulin injection, the blood glucose concentrations were measured against time and the AUC was calculated. (**d**) Serum insulin and leptin levels using ELISA. (**e**) Representative H&E staining and quantitative area of Langerhans islet cells in pancreatic sections. Scale bars, 100 µm. Significance was determined by one-way ANOVA. * *p* < 0.05 vs. low-fat diet (LFD)-fed mice. † *p* < 0.05 vs. high-fat diet (HFD)-fed mice.

**Figure 3 nutrients-14-05049-f003:**
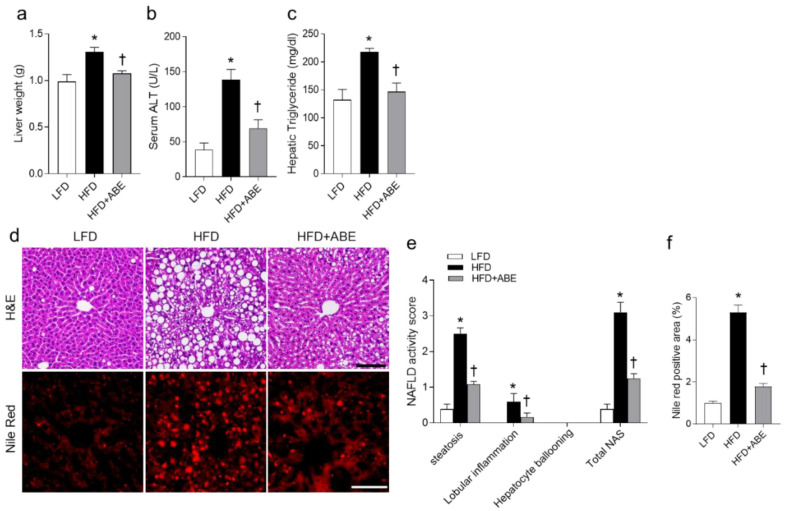
Effects of adzuki bean MY59 extract (ABE) on hepatic steatosis in HFD-fed mice. (**a**) Liver weight. (**b**) Serum alanine aminotransferase (ALT). (c) Hepatic triglyceride level. (**d**) Representative H&E and Nile Red staining in liver sections. Scale bars, 100 µm. (**e**) NAFLD activity score. (**f**) Quantification of Nile Red positive areas. Significance was determined by one-way ANOVA. * *p* < 0.05 vs. low-fat diet (LFD)-fed mice. † *p* < 0.05 vs. high-fat diet (HFD)-fed mice.

**Figure 4 nutrients-14-05049-f004:**
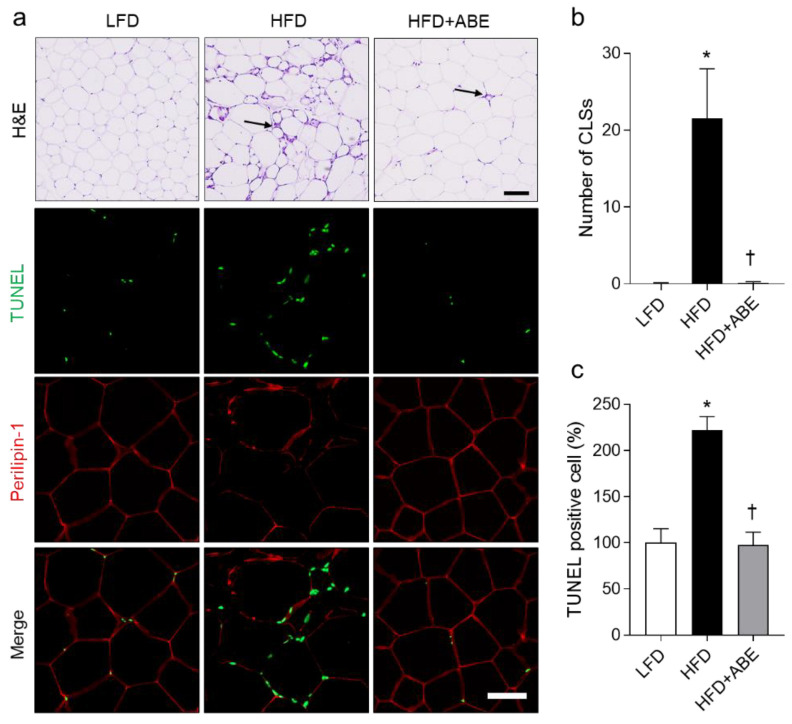
Effects of adzuki bean MY59 extract (ABE) on apoptotic adipocyte in HFD-fed mice. (**a**) Representative H&E and immunofluorescence staining of perilipin-1 with TUNEL in the epididymal fat pad sections. Arrow indicates crown like structure (CLS). Scale bar, 50 µm. (**b**) Quantification of CLSs from H&E-stained sections. (**c**) Quantification of TUNEL-positive cells from immunofluorescence-stained sections. Significance was determined by one-way ANOVA. * *p* < 0.05 vs. low-fat diet (LFD)-fed mice. † *p* < 0.05 vs. high-fat diet (HFD)-fed mice.

**Figure 5 nutrients-14-05049-f005:**
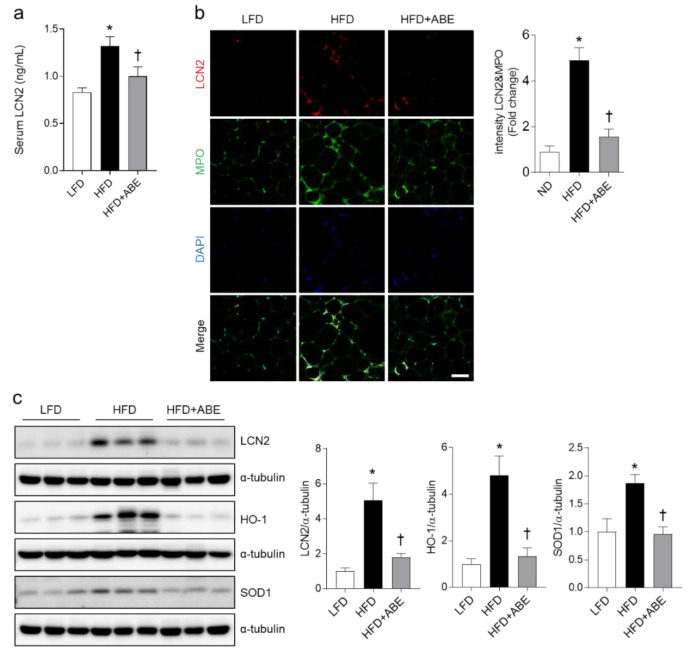
Effects of adzuki bean MY59 extract (ABE) on inflammation and oxidative stress in the adipose tissue of HFD-fed mice. (**a**) Serum lipocalin-2 (LCN2) using ELISA. (**b**) Representative images of double immunofluorescence staining of LCN2 and myeloperoxidase (MPO) in the epididymal fat pad sections. Nuclei were stained with 4′, 6-diamidino-2-phenylindole (DAPI). Quantification of co-localized LCN2 and MPO-immunostained density. Scale bar, 50 µm. (**c**) Western blot and quantitative analysis of LCN2, heme oxygenase-1 (HO-1), and superoxide dismutase1 (SOD1) protein in epididymal fat pad lysate. Protein levels were normalized to α-tubulin from the same immunoblot, respectably. Significance was determined by one-way ANOVA. * *p* < 0.05 vs. low-fat diet (LFD)-fed mice. † *p* < 0.05 vs. high-fat diet (HFD)-fed mice.

**Figure 6 nutrients-14-05049-f006:**
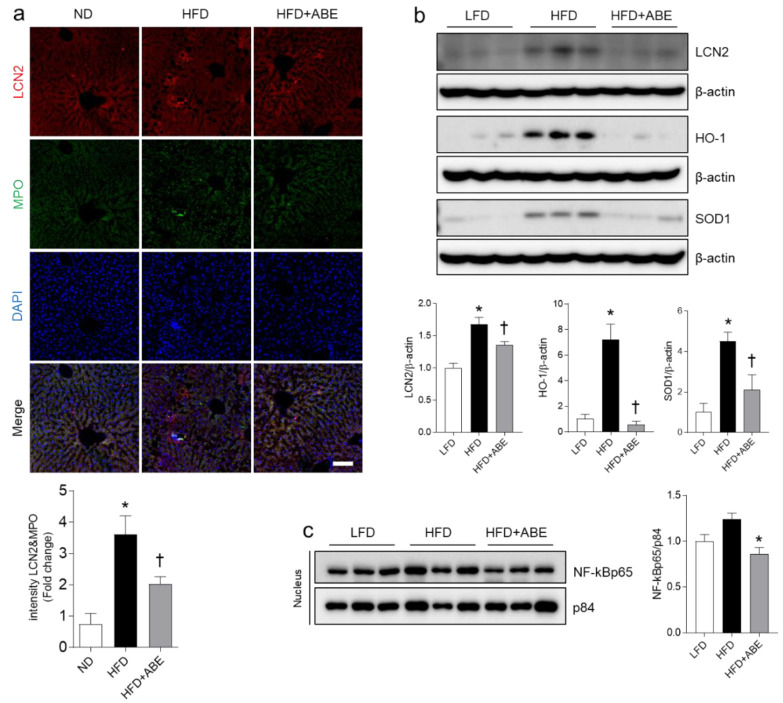
Effects of adzuki bean MY59 extract (ABE) on inflammation and oxidative stress in the liver of HFD-fed mice. (**a**) Representative images of double immunofluorescence staining of lipocalin-2 (LCN2) and myeloperoxidase (MPO) in the liver sections. Nuclei were stained with 4′, 6-diamidino-2-phenylindole (DAPI). Quantification of co-localized LCN2 and MPO-immunostained density. Scale bar, 50 µm. (**b**) Western blot and quantitative analysis of LCN2, heme oxygenase-1 (HO-1), and superoxide dismutase1 (SOD1) protein in the liver lysates. Protein levels were normalized to β-actin from the same immunoblot. (**c**) Western blot and quantitative analysis of NF-kBp65 in the nuclear fraction of liver tissue. Protein levels were normalized to p84 from the same immunoblot. Significance was determined by one-way ANOVA. * *p* < 0.05 vs. low-fat diet (LFD)-fed mice. † *p* < 0.05 vs. high-fat diet (HFD)-fed mice.

**Table 1 nutrients-14-05049-t001:** Inhibitory effects of adzuki bean extracts on α-glucosidase activities.

	α-Glucosidase Inhibitory Activity (IC_50,_ μg/mL)	
Bean Extracts and Compound	No Heat	Heat over 80 °C
*Vigna nakashimae*	9.7 ± 0.38	>500
Chungju-pat	>500	-
1st adzuki bean MY59 extract	6.4 ± 0.34	>500
2nd adzuki bean MY59 extract	1.9 ± 0.17	2.4 ± 0.16
Acarbose	140.5 ± 4.12	-

## Data Availability

The data presented in this study are available in this manuscript.

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
