# Peer review of "Adzuki Bean MY59 Extract Reduces Insulin Resistance and Hepatic Steatosis in High-Fat-Fed Mice via the Downregulation of Lipocalin-2"

_nutrients, 2022, doi:10.3390/nu14235049_

Round 1
Reviewer 1 Report
I am pleased to read this research results indicate that ABE feed- 24 ing could improve insulin resistance and hepatic steatosis by decreasing LCN2-mediated inflam- 25 mation and oxidative stress in HFD-fed mice, and Adzuki bean has this function.
Adzuki bean can preventing lipid accumulation and adipocytokine 286 production and improving glucose tolerance. it's very important.
I suggestion to add a picture of Adzuki bean in the paper, so that common people reading this paper can know exactly which plant it is.
Reviewer 2 Report
Thank you for the privilege of reviewing the manuscript on Adzuki bean extraxt effects on insulin resistance and hepatic steatosis in high-fat fed mice.
The article is very interesting and the study is performed correctly.
Please insert a paragraph on "limitations of study" at the end of discussion section.
